# Learning Latent Process from High-Dimensional Event Sequences via Efficient Sampling

**Qitian Wu**[1,2], **Zixuan Zhang**[1,2], **Xiaofeng Gao**[1,2]*, **Junchi Yan**[2,3], **Guihai Chen**[4]
[1]Shanghai Key Laboratory of Scalable Computing and Systems
[2]Department of Computer Science and Engineering, Shanghai Jiao Tong University
[3]MoE Key Lab of Artificial Intelligence, Shanghai Jiao Tong University
[4]State Key Labrotary of Novel Software Technology, Nanjing University
`{echo740, zzx_gongshi117}@sjtu.edu.cn, gao-xf@cs.sjtu.edu.cn`
`yanjunchi@sjtu.edu.cn, gchen@nju.edu.cn`

## Abstract

We target modeling latent dynamics in high-dimension marked event sequences without any prior knowledge about marker relations. Such problem has been rarely studied by previous works which would have fundamental difficulty to handle the arisen challenges: 1) the high-dimensional markers and unknown relation network among them pose intractable obstacles for modeling the latent dynamic process; 2) one observed event sequence may concurrently contain several different chains of interdependent events; 3) it is hard to well define the distance between two high-dimension event sequences. To these ends, in this paper, we propose a seminal adversarial imitation learning framework for high-dimension event sequence generation which could be decomposed into: 1) a *latent structural intensity model* that estimates the adjacent nodes without explicit networks and learns to capture the temporal dynamics in the latent space of markers over observed sequence; 2) an efficient *random walk based generation model* that aims at imitating the generation process of high-dimension event sequences from a bottom-up view; 3) a discriminator specified as a seq2seq network optimizing the rewards to help the generator output event sequences as real as possible. Experimental results on both synthetic and real-world datasets demonstrate that the proposed method could effectively detect the hidden network among markers and make decent prediction for future marked events, even when the number of markers scales to million level.

## 1 Introduction

Event sequence, consisting of a series of tuples $(time, marker)$ that records at which time which type of event takes place, could be a fine-grained representation [10] of temporal data that are pervasive in real-life applications. For example, one tweet or topic in social networks could give rise to huge number of forwarding behaviors, forming an information cascade. Such a cascade can be recorded by an event sequence composed of what time each retweet happens and who (a user) forwards the tweet, i.e., the marker. Another typical example is the POI route of a visitor in city, and the event sequence records when the person visits which POI and the POI is the marker. Also, there are cases where the markers contain compositional features, like job-hopping events in one period, where the event sequence records at which time *who* from *which* department of *which* company transfers to *which* department of *which* company. In this case, the marker contains five-dimension information. In the above examples, the number of markers could easily scale to an astronomical value when: 1) there are billions of users in one social network like Twitter; 2) there are a wealth of POIs in a big city; and

3) the compositional features stem from plenty of dimensions. In the literature, these event sequences with a huge number of event types are termed as *high-dimension (marked) event sequence* [6].

One problem for marked event sequence is to model the latent dynamic process from observed sequences. Such a latent process can be further decomposed into two mutually dependent processes: *temporal point process*, which captures the temporal dynamics between two adjacent events, and *relation network*, which denotes the dependencies among different markers. There are plenty of previous studies targeting the problem from different aspects. For temporal point process, a great number of works [3, 13, 15, 16, 28] attempt to model the intensify function from statistic views, and recent studies harness deep recurrent model [24], generative adversarial network [23] and reinforcement learning [19, 18] to learn the temporal process. These researches mainly focus on one-dimension event sequences where each event possesses the same marker. For marker relation modelling, several early studies [12, 27, 25] assume static correlation coefficients among markers and in some later works, the static coefficients are replaced by a series of parametric or non-parametric density functions [9, 11, 7]. Nevertheless, since these works need to learn dozens of parameters for each edge, which induces $O(n^2)$ parameter space, they are mostly limited in cases of multi-dimensional event sequences, where the number of markers is up to on hundred level.

There are a few existing studies that attempt to handle high-dimensional markers in one system. For instance, [8] targets information estimation in continuous-time diffusion network where each edge entails a transmission function. Several similar works like [2, 17] also focus on temporal point process in a huge diffusion network. However, they assume a given topology of the network, and differently in our work, the network of markers is unknown. Furthermore, [22] directly models the latent process from observed event sequences without the known network and tries to capture the dependencies among markers through temporal attention mechanism, which, nevertheless, could only implicitly reflect the relation network, while we aim at explicitly uncovering the hidden network with better interpretability. Moreover, the authors in [1] build a probabilistic model to uncover the time-varying networks of dependencies. By contrast, apart from network reconstruction, our paper also deals with the temporal dynamic process over the graph.

Learning latent process in high-dimension event sequences is highly intractable. Firstly, due to the huge number of markers, the unknown network could be pretty sparse, which makes previous methods assuming density function for each edge fail to work. The high-dimension markers also require a both effective and efficient representation. Secondly, one event sequence may consist of several different subsequences each of which entails a chain of interdependent event markers. In other words, two time-adjacent events in one sequence do not necessarily mean they possess dependency since the latter event may be caused by an earlier event. Such phenomenon makes the relations among events quite implicit. Thirdly, it is hard to quantify the discrepancy between two event sequences when events possess different markers. However, a proper loss function, which is the premise for decent model accuracy, highly requires a well-defined distance measure.

To these ends, in this paper, we propose a seminal adversarial imitation learning framework that aims at imitating the latent generation process of high-dimension event sequences. The main intuition behind the methodology is that if the model can generate event sequences close to the real ones, one can believe that the model has accurately captured the latent process. Specifically, the generator model contains two sub-modules: 1) a *latent structural intensity model*, which uses one marker's embedding feature to estimate a group of markers that are possibly its first-order neighbors and captures the temporal point process in the latent space of observed markers, and 2) an efficient *random walk based generation model*, which attempts to conduct a random walk on the local relation network of markers and generate the time and marker of next event based on the historical events. The special generator taking a bottom-up view for event generation with good interpretability could generalize to arbitrary cases without any parametric assumption, and can as well be efficiently implemented based on our theoretical insights. To detour the intractable distance measure for high-dimension event sequences, we design a seq2seq discriminator that maximizes reward on ground-truth event sequence (expert policy) and minimizes reward on generated one which will be further used to train the generator. To verify the model, we run experiments on two synthetic datasets and two real-world datasets. The empirical results show that the proposed model can give decent prediction for future events and network reconstruction, even when the number of markers scale to very high dimensions.

## 2 Methodology

**Preliminary for Temporal Point Process.** Event sequence can be modeled as a point process [4] where each new event's arrival time is treated as a random variable given the history of previous events. A common way to characterize a point process is via a conditional intensity function defined as: $\lambda(t|\mathcal{H}_t) = \frac{\mathbb{P}(N(t+dt)-N(t)=1|\mathcal{H}_t)}{dt}$, where $\mathcal{H}_t$ and $N(t)$ denote the history of previous events and number of events until time $t$, respectively. Then the arrival time of a new event would obey a density distribution $f(t|\mathcal{H}_t) = \lambda(t|\mathcal{H}_t)\exp(-\int_{t_n}^{t}\lambda(\tau|\mathcal{H}_t)d\tau)$, while the marker of the new event obeys a certain discrete distribution $p(m|\mathcal{H}_t)$.

**Notations and Problem Formulation.** Assume that a system has $M$ types of events, i.e. markers, denoted as $\mathcal{M} = \{m_i\}_{i=1}^{M}$, and $M$ can be arbitrarily large. There exists a hidden relation network $\mathcal{G} = (\mathcal{V}, \mathcal{E})$, where $\mathcal{V} = \mathcal{M}$ and $\mathcal{E} = \{c_{ij}\}_{M \times M}$ denotes a set of directed edges. Here $c_{ij} = 1$ indicates that marker $m_j$ is the descendant of marker $m_i$ (i.e., an event with marker $m_i$ could cause an event with marker $m_j$), and $c_{ij} = 0$ denotes independence between two markers. An event sequence $\mathcal{S}$ entails a series of events with time and marker, denoted as $\mathcal{S} = \{(t_k, m_{i_k})\}$ $(k = 0, 1, \cdots)$ where $t_k$ and $m_{i_k}$ denote time and marker of the $k$-th event, respectively, and $m_{i_k}$ is the descendant of one of previous markers $m_{i_n}$ where $0 \leq n < k$. Note that it is possible that an event is caused by more than one events before, and we only consider the first parent as the true parent [11, 9, 8]. We call event $(t_0, m_{i_0})$ as source event. The problem in this paper can be formulated as follows. Given observed event sequences $\{\mathcal{S}\}$, we aim at recovering the hidden relation network $\mathcal{G}$ and learning the latent process in event sequences, i.e., the conditional distribution $\mathbb{P}((t_{k+1}, m_{i_{k+1}})|\mathcal{H}_k)$ where $\mathcal{H}_k = \{(t_n, m_{i_n})\}_{n=0}^{k}$ denotes the history up to time $t_k$.

**Model Overview.** The fundamental idea of our methodology is to imitate the event generation process from a bottom-up view where the time and marker of each new event are sampled based on the history of previous events and the network. Such idea could be justified by the main intuition that the model conceivably succeeds to capture the latent process once it can generate event sequences which are close to the ground-truth ones. To achieve this goal, we build a framework named LANTERN (Learning Latent Process in High-Dimension Marked Event Sequences), shown in Fig. 1. We will go into the details in the following.

### 2.1 Generating High-Dimension Event Sequences

**Latent Structural Intensity Model.** For marker $m_i$, we use an $M$-dimension one-hot vector $\mathbf{v}_i$ to represent it. Then by multiplying an embedding matrix $\mathbf{W}_M \in \mathbb{R}^{D \times M}$, we can further encode each marker into a latent semantic space and obtain its representation $\mathbf{d}_i = \mathbf{W}_M \mathbf{v}_i$. The embedding matrix $\mathbf{W}_M$ is initially assigned with random values and will be updated in training so as to capture the similarity between markers on semantic level. Given the history of event sequence (up to time $t_k$) with the first $k+1$ events, i.e., $\mathcal{H}_k = \{(t_n, m_{i_n})\}_{n=0}^{k}$, we build an deep attentive intensity model to capture both the temporal dynamic and structural dependency in the event sequence.

For the $n$-th event, the marker $m_{i_n}$ corresponds to a $D$-dimension embedding vector $\mathbf{d}_{i_n}$. To obtain a consistent representation, we embed the continuous time by adopting a linear transformation $\mathbf{t}_n = \mathbf{w}_T t_n + \mathbf{b}_T$, where $\mathbf{w}_T, \mathbf{b}_T \in \mathbb{R}^{D \times 1}$ are two trainable vectors. Then we linearly add the embeddings of marker and time, $\mathbf{e}_n = \eta \cdot \mathbf{t}_n + \mathbf{d}_{i_n}$, to represent the $n$-th event, incorporating both temporal and structural information. Later, we define a $D$-dimension intensity function by attentively aggregating the representations of all previous events,

$$\mathbf{h}_n = MultiHeadAttn(\mathbf{e}_0, \mathbf{e}_1, \cdots, \mathbf{e}_k), n = 0, 1, \cdots, k, \tag{1}$$

where the $MultiHeadAttn(\cdot)$ is specified in Appendix A.

**Remark.** The equation (1) computes a $D$-dimension intensity function in the latent space of high-dimension markers. Compared with previous works that rely on a scalar intensity value for each dimension (specified by either statistic functions or deep models), our model possesses two advantages. Firstly, the marker embedding enables (1) to capture the structural proximity among markers in a latent space and the value of $\mathbf{h}_k$ implies the instantaneous arrival rate of new markers on semantic level. Such property enables our model to express more complex dynamics with efficiency, especially for high-dimension event sequences. Secondly, the time is encoded as vector representation, instead

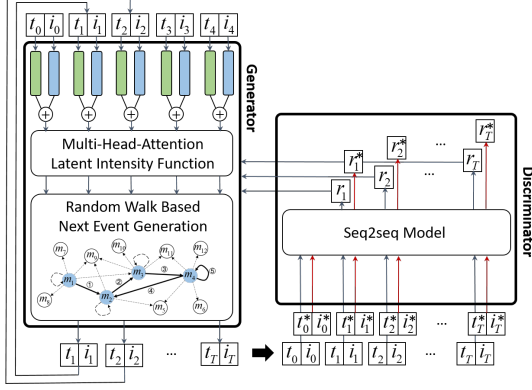

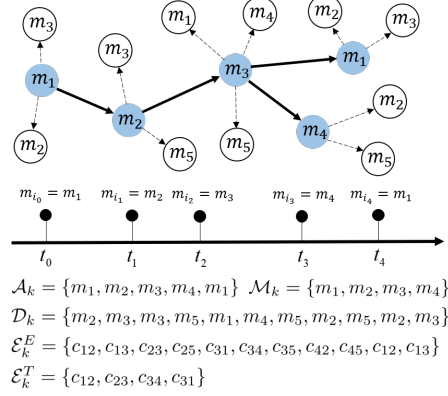

Figure 1: Framework of LANTERN: a generator leverages multi-head attention units to capture the intensify function in the latent space of markers and a random walk method to generate next event, and a discriminator aims at optimizing the reward for each sampling.

Figure 2: Local relation network of an event sequence (with $k = 4$). The blue nodes represent event markers existing in the sequence, and the white nodes belong to their causal descendants.

of directly concatenating as a scalar value with the marker embedding in previous works. Such setting is similar to the position embedding [20, 5] for sentence representation in NLP task, while the difference is that for event sequences we deal with continuous time, which is more fine-grained than discrete positions.

**Random Walk Based Next Event Generation.** Due to the causal-effect nature in event sequences, new event marker could only lie in the descendants of all existing markers. Use $\mathcal{M}_k = \cup_{n=0}^{k}\{m_{i_n}\}$ to denote the set of existing markers in $\mathcal{H}_k$. For $m_i \in \mathcal{M}_k$, its descendants in relation network could be estimated by attentively sampling over

$$p(m_j \in \mathcal{N}_i) = \frac{\exp(\mathbf{w}_C^\top[\mathbf{d}_j||\mathbf{d}_i])}{\sum_{u=1}^{M} \exp(\mathbf{w}_C^\top[\mathbf{d}_u||\mathbf{d}_i])}, \tag{2}$$

where $\mathcal{N}_i = \{m_j|c_{ij} = 1\}$ denotes the set of descendants of $m_i$ in $\mathcal{G}$. Such sampling method is inspired by graph attention network (GAT) [21], while the difference is that GAT aims at encoding a given network as feature vectors and on the contrary, our model uses the trainable embeddings of nodes to retrieve the network. The denominator of (2) requires embedding for each marker in the system, which poses high computational cost for training. Thus in practical implementation, we can fix all $p(m_j \in \mathcal{N}_i)$ during one epoch, and update the parameters when the epoch finishes. That could reduce the complexity as well as avoid high variance for event generation in different mini-batches.

The probability that the $n$-th event $(t_n, m_{i_n})$ in the history sequence $\mathcal{H}_k$ causes a new event with marker $m_j \in \mathcal{N}_{i_n}$ can be approximated by

$$p(m_j \in \overline{\mathcal{N}}_{i_n}|m_j \in \mathcal{N}_{i_n}) = \frac{\exp(\mathbf{w}_N^\top[\mathbf{h}_n||\mathbf{d}_j] + b_N)}{\sum_{m_u \in \mathcal{N}_{i_n}} \exp(\mathbf{w}_N^\top[\mathbf{h}_n||\mathbf{d}_u] + b_N)}, \tag{3}$$

where $\overline{\mathcal{N}}_{i_n}$ denotes true descendants of marker $m_{i_n}$ in the event sequence.

To sample new marker $m_{i_{k+1}}$, we design a random walk approach which interprets the generation process from a bottom-up view. Consider a multiset $\mathcal{A}_k$ (which allows one element appears multiple times) consisting of all existing markers in $\mathcal{H}_k$, and a multiset $\mathcal{D}_k$ containing the descendants of all existing markers. Besides, we define two another multisets: $\mathcal{E}_k^E = \{c_{i_nj}\}$ ($n = 1, \cdots, k$) which contains relation edges where $c_{i_nj}$ connects existing marker $m_{i_n} \in \mathcal{A}_k$ with its descendant $m_j \in \mathcal{N}_{i_n}$ given by sampling over (2) and $\mathcal{E}_k^T = \{c_{i_{n-1}i_n}\}$ ($n = 1, \cdots, k$) which contains true relation edges where $c_{i_{n-1}i_n}$ connects two event marker $m_{i_{n-1}}, m_{i_n} \in \mathcal{A}_k$ and $m_{i_n} \in \overline{\mathcal{N}}_{i_{n-1}}$. Define $\mathcal{V}_k = \mathcal{A}_k \cup \mathcal{D}_k$ and $\mathcal{E}_k = \mathcal{E}_k^E \cup \mathcal{E}_k^T$. Then $\mathcal{V}_k$ and $\mathcal{E}_k$ would induce a graph $\mathcal{G}_k = (\mathcal{V}_k, \mathcal{E}_k)$ which we call *local relation network*. Fig. 2 shows an example of local relation network for event sequences where the solid lines denote true relation edges. By definition, the new marker can be only sampled from $D_k$, i.e. the leaf nodes in $\mathcal{G}_k$.

---

**Algorithm 1:** Efficient Random Walk based Sampling for Generation of Next Event Marker

---

1 **INPUT:** $(t_0, m_{i_0})$, source event time and marker (which can be given or initially sampled from $\mathcal{M}$);
   $\mathcal{N}_i$, sampled descendants for each marker $m_i$, $i = 1, \cdot, M$.

2 $\mathcal{D}_0 \leftarrow \mathcal{N}_{i_0}$, set $\rho_0(m_j) = p(m_j \in \overline{\mathcal{N}}_{i_0} | m_j \in \mathcal{N}_{i_0})$ according to (1) and (3) for each $m_j \in \mathcal{N}_{i_0}$;

3 **for** $k = 1, \cdots, T$ **do**

4      Draw $m_{i_k}$ from $\mathcal{MN}(\rho_{k-1})$, and update $\mathcal{D}_k \leftarrow \mathcal{D}_{k-1} \cup \mathcal{N}_{i_k}$ // Assume $m_{i_n}$ is parent of $m_{i_k}$
        and we need to keep record of the parent of each $m_j \in \mathcal{D}_k$;

5      $b_k \leftarrow \rho_{k-1}(m_{i_n}) \cdot p(m_{i_k} \in \overline{\mathcal{N}}_{i_n} | m_i \in \mathcal{N}_{i_n})$, where $m_{i_k} \in \overline{\mathcal{N}}_v$;

6      $\rho_{k-1}(m_{i_k}) \leftarrow \rho_{k-1}(m_{i_k}) - b_k$;

7      **for** $m_i \in \mathcal{N}_{i_k}$ **do**

8          $\rho_k(m_i) \leftarrow b_k \cdot p(m_i \in \overline{\mathcal{N}}_{i_k} | m_i \in \mathcal{N}_{i_k})$;

9 **OUTPUT:** $\mathcal{S} = \{(t_k, m_{i_k})\}_{k=0}^{T}$, a generated event sequence.

---

For each $m_j \in \mathcal{D}_k$, use $\mathcal{P}_j^k$ to denote the path from $m_{i_0}$ to $m_j$ and $P_j^k = \{m_{u_n}\}_{n=0}^{N}$ contains each marker $m_{u_n}$ on the path where $m_{u_0} = m_{i_0}$ and $m_{u_N} = m_j$. (Note that here $N$ varies with different $j$ and we omit the subscript here to keep notation clean). Here, $\mathcal{P}_j^k$ possesses an important property based on the causal-effect nature of event sequences.

**Theorem 1.** *In local relation network $\mathcal{G}_k = (\mathcal{V}_k, \mathcal{E}_k)$, for any $m_j \in \mathcal{D}_k$, each path $P_j^k = \{m_{u_n}\}_{n=0}^{N}$ satisfies that for any $n$, $0 \leq n < N$, it holds $m_{u_n} \in \mathcal{A}_k$.*

Then we give our random walk based generation process for next event:

- **Marker Generation:** start with the source event marker $m_{i_0}$, and when the current move is from marker $m_{i_n}$ to $m_i$: if $m_i \in \mathcal{A}_k$, jump to the next marker $m_j \in \mathcal{N}_i$ with probability $p(m_j \in \overline{\mathcal{N}}_{i_n} | m_j \in \mathcal{N}_{i_n})$ given by (3); otherwise, i.e., $m_i \in \mathcal{D}_k$ stop and set $m_{i_{k+1}} = m_i$.

- **Time Estimation:** we estimate the time interval between next event and the $k$-th event as $\Delta t_{k+1} = \log(1 + \exp(\mathbf{W}_T' \mathbf{h}_n + \mathbf{b}_T'))$, where $\mathbf{h}_n$ is the intensity representation up to time $t_n$ and $(t_n, m_{i_n})$ is the $n$-th event in $\mathcal{H}_k$. Finally, $t_{k+1} = t_k + \Delta t_{k+1}$.

Theorem 1 guarantees the well-definedness of the above interpretable approach. However, its theoretical complexity is quadratic w.r.t the maximum length of event sequences. We further propose an equivalent sampling method that requires linear time complexity.

**Efficient Algorithm.** For each $m_j \in \mathcal{D}_k$, the path $P_j^k = \{m_{u_n}\}_{n=0}^{N}$ would induce a probability $p(P_j^k | \mathcal{G}_k) = \prod_{n=1}^{N} p(m_{u_n} \in \overline{\mathcal{N}}_{u_{n-1}} | m_{u_n} \in \mathcal{N}_{u_{n-1}})$. Then we can obtain the following theorem.

**Theorem 2.** *The random walk approach is equivalent to drawing a marker $m_j$ from $\mathcal{D}_k$ according to a multinomial distribution $\mathcal{MN}(\rho)$ where $\rho(m_j) = p(P_j^k | \mathcal{G}_k)$.*

Theorem 2 allows us to design an alternative sampling algorithm by iteratively using previous outcomes, which is shown in Alg. 1. We further show that the sampling method of Alg. 1 is well-defined and equivalent to the one in Theorem 2. Also, its complexity is linear w.r.t the sequence length. The proofs are in appendix B.

## 2.2 Training by Inverse Reinforcement Learning

**Optimization.** As discussed in previous subsection, the main goal of our model is to generate event sequences as real as possible. The generator can be treated as an agent who interacts with the environment and gives policy $\pi(a_k|s_k)$, where action $a_k = (t_k, m_{i_k})$ and state $s_k = \mathcal{H}_{k-1}$. Here $\pi(a_k|s_k) = \sum_{m_i \in \mathcal{M}_{k-1}} p(m_{i_k} \in \mathcal{N}_{m_i}) \cdot \rho_{k-1}(m_{i_k})$. The goal is to maximize the expectation of reward $r(\mathcal{S}) = r(a, s) = \sum_k \gamma^k r(a_k, s_k) = \sum_k \gamma^k r_k$, where $\gamma$ is a discount factor. Since to measure the discrepancy between two high-dimension event sequences is quite intractable, it is hard to determine a proper reward function. We thus turn to inverse reinforcement learning which

concurrently optimizes the reward function and policy network, and the objective can be written as

$$\min_\pi -H(\pi) + \max_r \mathbb{E}_{\pi_E}(r(\mathcal{S}^*)) - \mathbb{E}_\pi(r(\mathcal{S})), \tag{4}$$

where $\mathcal{S} = \{(t_k, m_{i_k})\}$ $(\mathcal{S}^* = \{(t_k^*, m_{i_k}^*)\})$ is the generated (ground-truth) event sequences given the same source event, $\pi_E$ is the expert policy that gives $\mathcal{S}^*$, and $H(\pi)$ denotes entropy of policy.

We proceed to adopt the GAIL [14] framework to learn the reward function by considering a discriminator $D_w : \mathcal{S} \to [0,1]^T$, which is parametrized by $w$ and maps an event sequence to a sequence of rewards $\{r_k\}_{k=1}^T$ in the range $[0,1]$. Then the gradient for the discriminator is given by

$$\mathbb{E}_\pi[\nabla_w \log D_w(\mathcal{S})] + \mathbb{E}_{\pi_E}[\nabla_w \log(1 - D_w(\mathcal{S}^*))]$$
$$\approx \frac{1}{B} \sum_{b=1}^B \sum_{k=1}^T \nabla_w \log d_k(\mathcal{S}_b; w) + \sum_{k=1}^T \nabla_w \log(1 - d_k(\mathcal{S}_b^*; w)), \tag{5}$$

where $d_k(\mathcal{S}; w) = r_k$ is the $k$-th output of $D_w(\mathcal{S})$ and we sample $B$ generated sequences $\{\mathcal{S}_b\}_{b=1}^B$ to approximate the expectation. Then we give the policy gradient for the generator with parameter set $\theta$:

$$\mathbb{E}_{\pi_\theta}[\nabla_\theta \log \pi(a|s) \log D_w(\mathcal{S})] - \lambda \nabla_\theta H(\pi)$$
$$\approx \frac{1}{B} \sum_{b=1}^B \sum_{k=1}^T \gamma^k \nabla_\theta \log \pi(a_k|s_k) \log d_k(\mathcal{S}_b; w)) - \lambda \sum_{k=1}^T \nabla_\theta \log \pi(a_k|s_k) Q_{log}(a, s), \tag{6}$$

where $Q_{log}(\overline{a}, \overline{s}) = \mathbb{E}_{\pi_\theta}(-\log \pi_\theta(a|s)|s_0 = \overline{s}, a_0 = \overline{a})$.

The training algorithm is given by Alg. 2 in Appendix D.

**Ingredients of Discriminator.** We harness a sequence-to-sequence model to implement the discriminator model $D_w : \mathcal{S} \to [0,1]^T$. Given event sequence $\mathcal{S} = \{(t_k, m_{i_k})\}_{k=0}^T$ with event embedding $\mathbf{e}_0, \mathbf{e}_1, \cdots, \mathbf{e}_T$, we have

$$\mathbf{a}_k = MultiHeadAttn(\mathbf{e}_0, \mathbf{e}_1, \cdots, \mathbf{e}_T),$$
$$r_k = sigmoid(\mathbf{W}_D \mathbf{a}_k + b_D), k = 1, \cdots, T, \tag{7}$$

where $\mathbf{W}_D \in \mathbb{R}^{D \times 1}$ and $b_D$ is a scalar.

# 3  Experiments

We apply our model LANTERN to two synthetic datasets and two real-world datasets in order to verify its effectiveness in modeling high-dimension event sequences. The codes are released at https://github.com/zhangzx-sjtu/LANTERN-NeurIPS-2019.

**Synthetic Data Generation.** We generate two networks, a small one with 1000 nodes and a large one with 100,000 nodes, and the directed edges are sampled by a Bernoulli distribution with $p = 5 \times 10^{-3}$ for the small network and $p = 3 \times 10^{-5}$ for the large network. The nodes in network are treated as markers. Each edge $c_{ij}$ corresponds to a Rayleigh distribution $f_{ij}(t|a,b) = \frac{2}{t-a}(\frac{t-a}{b})^2 \exp(-(\frac{t-a}{b})^2)$, $t \geq a$. We basically set $a = 0$ and $b = 1$. Then we generate event sequences in this way: 1) randomly select a node as marker of source event and set the time of source event as 0; 2) for each sampled marker $i$, sample the time of next event with marker $j$ which is the descendant of marker $i$ in the network according to $f_{ij}(t)$ and pick the event with smallest time as new sampled event. The whole process repeats till the time exceeds a global time window $T^c$. If a sampled event marker has more than one parents, we use the smallest sampled time as the true sampled time of event. We repeat the above process and generate 10,000 event sequences for the small network and 100,000 event sequences for the large network. We call the dataset with small network as Syn-Small and the dataset with large network as Syn-Large.

**Real-World Data Information.** We also use two real-world datasets in our experiment. Firstly, MemeTracker dataset [11] contains hyperlinks between articles and records information flow from one site to another. In this setting, each site plays as a marker and each article would generate an information cascade which can be treated as an event sequence. The hyperlinks represent the relation

Table 1: Results for network reconstruction. We compare the estimated edges and ground-truth edges, and statistic precision (PRE), recall (REC) and F1 score (F1). For LANTERN, LANTERN-RNN and LANTERN-PR, we use the edges with top $K$ probabilities given by (2) for one marker as estimated edges and consider three different settings of $K$. In Syn-Small, Syn-Large, we set $K_1 = 3$, $K_2 = 4$, $K_3 = 5$; in Memetracker and Weibo, we consider $K_1 = 25$, $K_2 = 30$, $K_3 = 35$.

| Methods | Syn-Small | | | Syn-Large | | | MemeTracker | | | Weibo | | |
|---|---|---|---|---|---|---|---|---|---|---|---|---|
| | PRE | REC | F1 | PRE | REC | F1 | PRE | REC | F1 | PRE | REC | F1 |
| NETRATE | 0.4983 | 0.3986 | 0.4429 | - | - | - | 0.5665 | 0.2447 | 0.3418 | - | - | - |
| KernelCascade | 0.4975 | 0.3980 | 0.4422 | - | - | - | 0.5364 | 0.2897 | 0.3762 | - | - | - |
| LTN-PR ($K_1$) | 0.5899 | 0.3539 | 0.4424 | 0.4740 | 0.4740 | 0.4740 | 0.4973 | 0.3357 | 0.4009 | 0.3824 | 0.3524 | 0.3654 |
| LTN-PR ($K_2$) | 0.5856 | 0.4685 | 0.5205 | 0.4987 | 0.4987 | 0.4987 | 0.4637 | 0.3756 | 0.4150 | 0.3560 | 0.3864 | 0.3692 |
| LTN-PR ($K_3$) | 0.5823 | 0.5823 | **0.5823** | 0.3984 | 0.6640 | 0.4980 | 0.4336 | 0.4098 | 0.4214 | 0.3302 | 0.3717 | 0.3484 |
| LTN-RNN ($K_1$) | 0.4476 | 0.2686 | 0.3357 | 0.6523 | 0.3914 | 0.4892 | 0.4998 | 0.3374 | 0.4028 | 0.5706 | 0.5274 | 0.5462 |
| LTN-RNN ($K_2$) | 0.4718 | 0.3774 | 0.4194 | 0.4980 | 0.6640 | 0.5691 | 0.4653 | 0.3769 | 0.4165 | 0.5417 | 0.5910 | 0.5631 |
| LTN-RNN ($K_3$) | 0.4888 | 0.4888 | 0.4888 | 0.4976 | 0.8293 | 0.6220 | 0.4352 | 0.4113 | 0.4211 | 0.5306 | 0.5966 | 0.5596 |
| LANTERN ($K_1$) | 0.5758 | 0.3455 | 0.4318 | 0.4833 | 0.4833 | 0.4833 | 0.4987 | 0.3367 | 0.4020 | 0.5726 | 0.5295 | 0.5483 |
| LANTERN ($K_2$) | 0.5742 | 0.4594 | 0.5104 | 0.5000 | 0.6667 | 0.5714 | 0.4651 | 0.3767 | 0.4163 | 0.5448 | 0.5944 | **0.5663** |
| LANTERN ($K_3$) | 0.5733 | 0.5733 | 0.5733 | 0.4952 | 0.8483 | **0.6253** | 0.4354 | 0.4114 | **0.4230** | 0.5320 | 0.5982 | 0.5611 |

network among markers. We filter a network of top 583 sites with 6700 cascades. The MemeTracker dataset is used to compare our model with some previous methods which focus on learning the network and temporal process in event sequences with hundreds of markers. Besides, we consider a large-scale dataset Weibo [26] which records the resharing of posts among $1,787,443$ users with $413,503,687$ following edges. Here each user corresponds to an event marker and every resharing behavior of user can be seen as an event, so the cascades of resharing would form an high-dimension event sequence. We extract $10^5$ users with $2531525$ edges and $10^5$ cascades to evaluate our model in modeling high-dimension event sequences.

**Competitors and Baselines.** We compare our model with two previous methods, NETRATE [11] and KernelCascade [9], which attempt to learn the heterogeneous network and the temporal process from event sequences by learning a transmission density function for each edge. Since their huge parameter size limits the scalability to very high-dimension markers, we only apply them to our small synthetic dataset and MemeTracker dataset. Besides, we consider two simplified versions of LANTERN as ablation study: LANTERN-RNN which replaces the multi-head attention mechanism by RNN structure, and LANTERN-PR which removes the discriminator and uses a heuristic reward function as training signal for generator. We compare our model with them in four datasets to study the effectiveness of attention mechanism and inverse reinforcement learning. For each method, we run five times and report the average values in this paper. All the improvements in this paper are significant according to the Wilcoxon signed-rank test on $5\%$ confidence level. The implementation details for baselines and hyper-parameter settings are presented in Appendix C.

**Event Prediction.** We use our model to predict the time and marker of next event given part of observed sequence, and use MSE and accuracy to evaluate the performance of time and marker prediction, respectively. The results of all methods are shown in Fig. 3. As we can see, in MemeTracker and Syn-SMALL, KernelCascade slightly outperforms NETRATE for both time and marker prediction, while our model LANTERN achieves great improvement over two competitors, especially when given very few observed events. LANTERN-RNN performs better compared with LANTERN-PR in small datasets. However, when the dimension of markers is extremely large, the performance of LANTERN-RNN bears a considerable decline probably due to the limited capacity of RNN architecture to capture the high-variational relations between high-dimensional event markers. In four datasets, LANTERN-PR is generally inferior to LANTERN for both time and marker prediction. The possible reason is that the heuristic reward function cannot well characterize the discrepancy between event sequences and may provide unreliable training signals.

**Network Reconstruction.** We also leverage the model to reconstruct the network topology and use precision, recall and F1 score as metrics. The results are shown in Table 1 where we shorten LANTERN-RRN and LANTERN-PR as LTN-RNN and LTN-PR, respectively. As shown in Table 1, LANTERN could give the best reconstruction F1 score among all baselines, and achieve averagely $14.9\%$ improvement over the better one of NETRATE and KernelCascade. Also, LANTERN outperforms LANTERN-RNN, which indicates that the multi-head attention network could better capture the latent structural proximity among markers in event sequences.

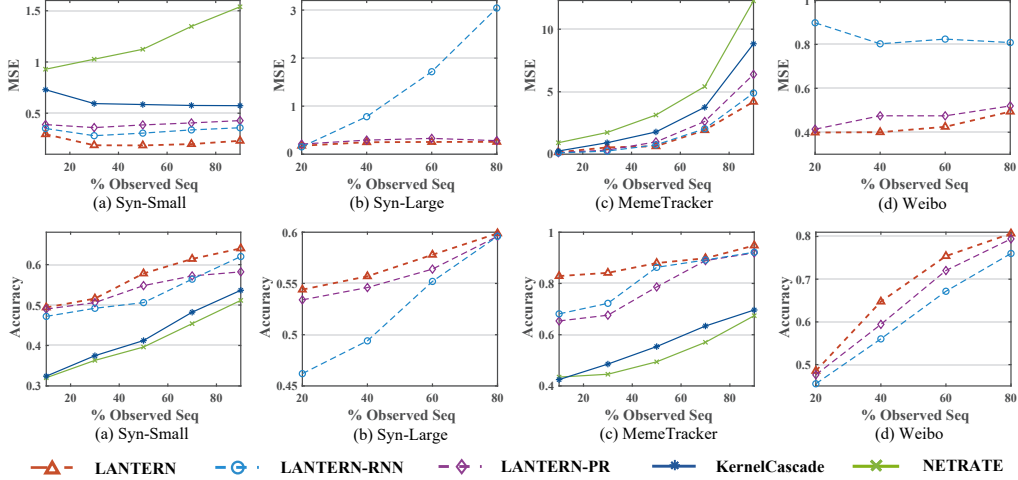

Figure 3: Experimental results for time and marker prediction in four datasets. We truncate a certain ratio of an event sequence as observed information and aim at predicting the time and marker of next event. The figures show the prediction performance under different observed ratios.

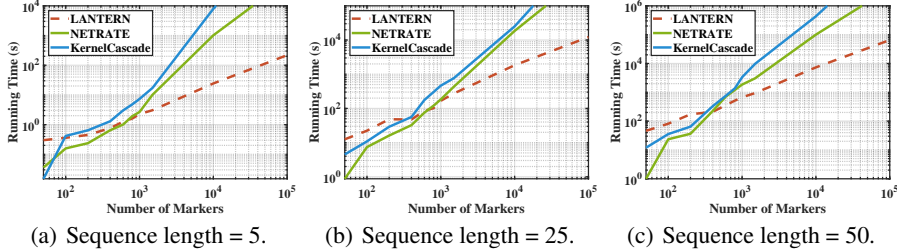

Figure 4: Scalability test in synthetic dataset. We change marker number from 100 to 100,000 and report running time of LANTERN, NETRATE and KernelCascade with different sequence lengths.

**Scalability.** We also test our model under different numbers of markers and sequence lengths, and present the results in Fig. 3. The experiments are deployed on Nvidia Tesla K80 GPUs with 12G memory and we statistic the running time to discuss the model scalability. It shows that with the marker number increasing from 100 to 100,000, the running time of LANTERN increases in a linear manner, while the trends of two other methods behave almost in an exponential way. When the system has a huge number of markers (like on million level), LANTERN is still effective with good scalability, but NETRATE and KernelCascade would be too time-consuming due to the fact that they need to optimize a transmission density function for each edge in the network, which induces at least quadratic parameter space in terms of marker number.

## 4 Conclusion

In this paper, we focus on learning both the hidden relation network and temporal point process in high-dimension marked event sequences, which has rarely been studied and poses some intractable challenges for previous approaches. To solve the problem, we firstly build a generator model that takes a bottom-up view to imitate the generation process of event sequences. The generator model considers each marker as an embedding vector, uses graph-based attentive estimation for network reconstruction, and entails a latent structural intensity function to capture the temporal point process in the latant space of markers over the sequence. Then we design an interpretable and efficient random walk based sampling approach to generate the next event. To overcome the difficulty of measuring the discrepancy between high-dimension event sequences, we use inverse reinforcement learning to optimize the reward function for event generation. Extensive experiments on both synthetic and (large-scale) real-world datasets demonstrate that our model could give superior prediction for future events as well as reconstruct the hidden network. Also, scalability tests show that the model can tackle event sequences with huge number of markers.

## 5 Acknowledgement

This work was supported by the National Key RD Program of China [2018YFB1004703]; the National Natural Science Foundation of China [61872238, 61672353, 61972250]; the Shanghai Science and Technology Fund [17510740200]; the CCF-Huawei Database System Innovation Research Plan [CCF-Huawei DBIR2019002A]; the Huawei Innovation Research Program [HO2018085286]; the State Key Laboratory of Air Traffic Management System and Technology [SKLATM20180X], and the Tencent Social Ads Rhino-Bird Focused Research Program.

## Footnotes

*Xiaofeng Gao is corresponding author.

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
