[Supplementary Material]

# A Details of Multi-Head Attention Unit

The $MultiHeadAttn(\cdot)$ function takes $k+1$ vectors $(\mathbf{e}_0, \mathbf{e}_1, \cdots, \mathbf{e}_k)$ as input and output $k+1$ vectors. It contains $L$ attention units and each of them independently aggregates the input vectors and obtain

$$\mathbf{h}_n^l = \sum_{j=0}^{n} \alpha_{jn}^l \mathbf{W}_V \mathbf{e}_j,$$

where $\alpha_{jn}^l$ is the attention weights from the $j$-th element to the $n$-th element given by the $l$-th attention head and

$$\alpha_{jn}^l = \frac{\exp((\mathbf{W}_K \mathbf{e}_j)^\top \mathbf{W}_Q \mathbf{e}_n)}{\sum_{i=0}^{n} \exp((\mathbf{W}_K \mathbf{e}_i)^\top \mathbf{W}_Q \mathbf{e}_n)},$$

where $\mathbf{W}_V \in \mathbb{R}^{D \times D}$, $\mathbf{W}_K \in \mathbb{R}^{D \times D}$ and $\mathbf{W}_Q \in \mathbb{R}^{D \times D}$ are three transformation matrices for value, key and query, respectively. Then the final representation $\mathbf{h}_n$ could be given by combining the outputs of each single-head attention function,

$$\mathbf{h}_n^\top = \mathbf{W}_O[(\mathbf{h}_n^1)^\top, \cdots, (\mathbf{h}_n^L)^\top], n = 0, 1, \cdots, k,$$

where $\mathbf{W}_O \in \mathbb{R}^{S \times L}$ is a matrix and $S = D \cdot L$. Here, the $n$-th output representation only takes the first $n+1$ vectors as input and masks other input vectors, which could guarantee the causality property in our setting.

# B Proofs in Section 2

## B.1 Proof for Theorem 2

The proof is by construction. We know that each random walk would realize a path from the source event marker to the sampled marker. Consider a probability space $(\Omega, \mathcal{F}, P)$ where $\Omega$ denotes a sample space including all possible outcomes, $\mathcal{F}$ is an event space and $P$ represents probability assignment of all events. In the random walk experiment, $\Omega = \cup_{m_j \in \mathcal{D}_k} \mathcal{P}_j^k = \{P_{j,c}^k | P_{j,c}^k \in \mathcal{P}_j^k, m_j \in \mathcal{D}_k\}$ contains all possible paths from the source event marker to any marker that is likely to be sampled. Then each outcome $P_{j,c}^k$ induces a probability

$$p(P_{j,c}^k | \mathcal{G}_k) = \prod_{n=1}^{N} p(m_{u_n} \in \overline{\mathcal{N}}_{u_{n-1}} | m_{u_n} \in \mathcal{N}_{u_{n-1}}).$$

We further consider a random event that $m_j$ is sampled by the random walk approach. Obviously, such an event entails a series of experiment outcomes specified by all possible paths in $\mathcal{P}_j^k$. According to addition principle, we arrive at $p(m_j) = \sum_c p(P_{j,c}^k | \mathcal{G}_k)$.

Since the random walk process would stop if and only if it reaches a marker $m_j \in \mathcal{D}_k$, we have $\sum_{m_j \in \mathcal{D}_k} p(m_j) = 1$. Hence we can construct another probability space $(\Omega', \mathcal{F}', P')$, where $\Omega' = \{m_j | m_j \in \mathcal{D}_k\}$ and $p(m_j) = \sum_c p(P_{j,c}^k | \mathcal{G}_k)$, which exactly characterizes the experiment where we draw a marker from $\mathcal{D}_k$ with multinomial distribution.

## B.2 Well-Definedness of Alg. 1

To guarantee the well-definedness, we need to verify the fact that in the $k$-th step, the sum of probabilities for all possible sampled markers equal to one, under the settings of Alg. 1. We rely on mathematical induction to finish the proof. In the 0-th step, only the causal descendants of the source event marker could be possibly sampled, so the argument is obviously true with

$$\sum_{m_j \in \mathcal{N}_{i_0}} \rho_0(m_j) = \sum_{m_j \in \mathcal{N}_{i_0}} p(m_j \in \overline{\mathcal{N}}_{i_0} | m_j \in \mathcal{N}_{i_0}) = 1,$$

where the last equality is obvious based on the softmax function in (3).

Assume that in the $(k-1)$-th step $(k = 1, 2, \cdots)$, we have

$$\sum_{m_j \in \mathcal{D}_{k-1}} \rho_{k-1}(m_j) = 1.$$

Then in the $k$-th step, assume $m_{i_k}$ to be the new sampled event and the true causal descendent of $m_v$ (i.e., $m_{i_k} \in \overline{\mathcal{N}}_v$), then we have

$$\sum_{m_j \in \mathcal{D}_k} \rho_k(m_j) = \sum_{m_j \in \mathcal{D}_k} \rho_{k-1}(m_j) - \rho_{k-1}(m_v) \cdot p(m_{i_k} \in \overline{\mathcal{N}}_v | m_i \in \mathcal{N}_v)$$

$$+ \sum_{\substack{m_i \in \mathcal{N}_{i_k} \\ m_i \notin \mathcal{D}_{k-1}}} (\rho_k(m_i) - \rho_{k-1}(m_i)) + \sum_{\substack{m_i \in \mathcal{N}_{i_k} \\ m_i \in \mathcal{D}_{k-1}}} \rho_k(m_i)$$

$$= \sum_{m_j \in \mathcal{D}_k} \rho_{k-1}(m_j) - \rho_{k-1}(m_v) \cdot p(m_{i_k} \in \overline{\mathcal{N}}_v | m_i \in \mathcal{N}_v)$$

$$+ \sum_{\substack{m_i \in \mathcal{N}_{i_k} \\ m_i \notin \mathcal{D}_{k-1}}} b_k \cdot p(m_i \in \overline{\mathcal{N}}_{i_k} | m_i \in \mathcal{N}_{i_k}) + \sum_{\substack{m_i \in \mathcal{N}_{i_k} \\ m_i \in \mathcal{D}_{k-1}}} b_k \cdot p(m_i \in \overline{\mathcal{N}}_{i_k} | m_i \in \mathcal{N}_{i_k})$$

$$= \sum_{m_j \in \mathcal{D}_k} \rho_{k-1}(m_j) - \rho_{k-1}(m_v) \cdot p(m_{i_k} \in \overline{\mathcal{N}}_v | m_i \in \mathcal{N}_v)$$

$$+ \sum_{m_i \in \mathcal{N}_{i_k}} b_k \cdot p(m_i \in \overline{\mathcal{N}}_{i_k} | m_i \in \mathcal{N}_{i_k})$$

$$= \sum_{m_j \in \mathcal{D}_k} \rho_{k-1}(m_j) - \rho_{k-1}(m_v) \cdot p(m_{i_k} \in \overline{\mathcal{N}}_v | m_i \in \mathcal{N}_v) + b_k$$

$$= \sum_{m_j \in \mathcal{D}_k} \rho_{k-1}(m_j).$$

Thus we have $\sum_{m_j \in \mathcal{D}_k} \rho_k(m_j) = 1$ and conclude the proof.

### B.3 Equivalence of Alg. 1 to Random Walk Sampling

According to Theorem 2, we only need to prove that through Alg. 1, for any $m_i \in \mathcal{D}_k$, $\rho_k(m_i) = \sum_c p(P_{i,c}^k | \mathcal{G}_k)$, which is true for $k = 0, 1, \cdots, T$. We rely on strong mathematical induction to conduct the proof. For $k = 0$, the argument is obvious true according to Alg. 1. Assume that for $k' = 0, \cdots, k-1$ ($k = 1, \cdots, T$), we have the following property: for any $m_j \in \mathcal{D}_{k'}$, $\rho_k(m_i) = \sum_c p(P_{i,c}^{k'} | \mathcal{G}_{k'})$. Then we need to prove the argument is true for $k$-th step.

For $m_i \notin \mathcal{N}_{i_k}$, we have $\rho_k(m_i) = \rho_{k-1}(m_i) = \sum_c p(P_{i,c}^{k-1} | \mathcal{G}_{k-1}) = \sum_c p(P_{i,c}^k | \mathcal{G}_k)$.

For $m_i \in \mathcal{N}_{i_k}$, consider two cases. If $m_i \in \mathcal{D}_{k-1}$, then

$$\rho_k(m_i) = \rho_{k-1}(m_i) + b_k \cdot p(m_i \in \overline{\mathcal{N}}_{i_k} | m_i \in \mathcal{N}_{i_k})$$

$$= \sum_c p(P_{i,c}^{k-1} | \mathcal{G}_{k-1}) + \rho_{k-1}(m_v) \cdot p(m_{i_k} \in \overline{\mathcal{N}}_v | m_i \in \mathcal{N}_v) \cdot p(m_i \in \overline{\mathcal{N}}_{i_k} | m_i \in \mathcal{N}_{i_k})$$

$$= \sum_c p(P_{i,c}^{k-1} | \mathcal{G}_{k-1}) + \sum_c p(P_{v,c}^{k-1} | \mathcal{G}_k) \cdot p(m_{i_k} \in \overline{\mathcal{N}}_v | m_i \in \mathcal{N}_v) \cdot p(m_i \in \overline{\mathcal{N}}_{i_k} | m_i \in \mathcal{N}_{i_k})$$

$$= \sum_c p(P_{i,c}^k | \mathcal{G}_k),$$

where the last equality is true due to the fact $\mathcal{P}_i^k = \mathcal{P}_i^{k-1} \cup \mathcal{P}_v^{k-1}$. If $m_i \notin \mathcal{D}_{k-1}$, then we have

$$\rho_k(m_i) = \rho_{k-1}(m_v) \cdot p(m_{i_k} \in \overline{\mathcal{N}}_v | m_i \in \mathcal{N}_v) \cdot p(m_i \in \overline{\mathcal{N}}_{i_k} | m_i \in \mathcal{N}_{i_k})$$

$$= \sum_c p(P_{v,c}^{k-1} | \mathcal{G}_k) \cdot p(m_{i_k} \in \overline{\mathcal{N}}_v | m_i \in \mathcal{N}_v) \cdot p(m_i \in \overline{\mathcal{N}}_{i_k} | m_i \in \mathcal{N}_{i_k})$$

$$= \sum_c p(P_{i,c}^k | \mathcal{G}_k),$$

where the last equality is based on the fact $\mathcal{P}_i^k = \mathcal{P}_v^{k-1}$. We conclude the proof.

## C Implementation Details

### C.1 Details of LANTERN-RNN

In LANTERN-RNN, we replace the multi-head attention unit by RNN structure to model the intensity function. Specifically,
$$\mathbf{h}_n = \phi(\mathbf{A}\mathbf{e}_n + \mathbf{B}\mathbf{h}_{n-1} + \mathbf{b}),$$

where $\mathbf{A}, B \in \mathbb{R}^{D \times D}$, $b \in \mathbf{R}^{D \times 1}$ and $\phi$ is an activation function.

## C.2    Details of LANTERN-PR

In LANTERN-PR, we directly use a pre-defined reward function for generated event sequence as training signal of the generator. Assume the generated and ground-truth event sequences as $\mathcal{S} = \{(t_k, m_{i_k})\}$ and $\mathcal{S}^* = \{(t_k^*, m_{i_k}^*)\}$, respectively, and we define the reward as

$$r_k = C - \|t_k - t_k^*\|^2 + \delta(m_{i_k} = m_{i_k}^*),$$

where $\delta(s)$ is an indicator function which returns 1 if $s$ is true and 0 otherwise, and $C$ is a constant which guarantees positive reward value.

## C.3    Hyper-Parameter Settings

The settings for hyper-parameters are as follows: embedding dimension $D = 10$ for Syn-Small, $D = 16$ for Syn-Large, $D = 8$ for MemeTracker, $D = 16$ for Weibo; causal descendant sample size $K = 3$ for Syn-Small and Syn-Large, $K = 5$ for MemeTracker, $K = 20$ for Weibo; number of attention heads $L = 4$; regularization coefficient $\lambda = 0.001$; time embedding weight $\eta = 0.3$; discount factor $\gamma = 0.99$; Adam parameters $\alpha = 10^{-7}$, $\beta_1 = 0.9$, $\beta_2 = 0.99$ for generator, and $\alpha = 10^{-5}$, $\beta_1 = 0.9$, $\beta_2 = 0.99$ for discriminator.

# D    Model Training Algorithm

---

**Algorithm 2:** Training Algorithm for LANTERN

---

1 **INPUT:** $\{\mathcal{S}^*\}$, ground-truth event sequences. $\mathcal{M}$, marker set. Initialized discriminator parameter $w^0$, generator parameter $\theta^0$. Adam hyper-parameter $\alpha, \beta_1, \beta_2$.

2 **for** $k = 1, \cdots, n_{step}$ **do**

3 $\quad$ Sample $B$ observed event sequences $\{\mathcal{S}_b^*\}_{b=1}^B$ from $\{\mathcal{S}^*\}$;

4 $\quad$ Generate event sequence $\mathcal{S}_b$ given by the source event in $\mathcal{S}_b^*$ by Alg. 1;

5 $\quad$ Update generator parameter from $\theta^k$ to $\theta^{k+1}$ using (6);

6 $\quad$ $\Delta L_D \leftarrow \frac{1}{B} \sum_{b=1}^{B} \sum_{k=1}^{T} \nabla_w \log d_k(\mathcal{S}_b; w) + \sum_{k=1}^{T} \nabla_w \log(1 - d_k(\mathcal{S}_b^*; w))$;

7 $\quad$ $w^{k+1} \leftarrow Adam(\Delta L_D, w^k, \alpha, \beta_1, \beta_2)$;

8 $\quad$ $\Delta L_G \leftarrow \frac{1}{B} \sum_{b=1}^{B} \sum_{k=1}^{T} \gamma^k \nabla_\theta \log \pi(a_k|s_k) \log d_k(\mathcal{S}_b; w)) - \lambda \sum_{k=1}^{T} \nabla_\theta \log \pi(a_k|s_k) Q_{log}(a, s)$;

9 $\quad$ $\theta^{k+1} \leftarrow Adam(\Delta L_G, \theta^k, \alpha, \beta_1, \beta_2)$;

10 **OUTPUT:** discriminator parameter $w$, generator parameter $\theta$.

---