[Reviews · NeurIPS 2019]

Reviewer 1



This paper addresses the problem of learning a sequence model for marked temporal point processes. The problem has been studied in the literature and is of interest to the community. However, the paper is not written clearly and lacks coherence, there are many typos, and the authors use a language that is not appropriate for a scientific paper. One of my criticisms of the paper is the use of "causal-graphs" in the paper. The term causal generally refers to causal statements which answer counter-factual statements of the form "what if ...", the authors use this term extensively to describe the graph which they recover using their method, while the recovered graph may not be the causal graph in the sense often understood by the community. My other criticism of the paper is lack of coherence between sections of the paper. This makes understanding the motivation behind every section very difficult. In its current form, it is very difficult to follow the logic behind each step and hence understanding the paper.

Reviewer 2



The paper presents several fresh ideas and techniques for leanring high-dimensional event sequence in the continuous time space, which is challenging and still remains open. Specifically, the authors devise a random walk based approach among the local causal graph of markers, for high-dimensional event generation with a linear complexity efficient implementation, after a rigrous theoretical study. Meanwhile, a latent intensity model is devised without explicitly defining the network structure. This model can be also useful for high-dimensional marker space, as it instead model the latent space rather than the raw marker space which can be intractable. The distance between two high-dimensional event sequences is modeled via adversarial imitation learning, which makes the distance computing more tractable and flexible. Overall the paper is well written and the technical novelty is solid. The experiments are comprehensive.

Reviewer 3



This paper targets modeling the temporal dynamics in high-dimension marked event sequences without any given causal network of markers. This is a difficult problem and rarely studies in previous studies. The authors propose a seminal adversarial imitation learning framework which consists of a generator model that takes a bottom-up view to imitate the generation process of event sequences and an interpretable and efficient random walk based sampling approach to generate the next event. The methods are novel and have good intuition. The paper is well written and easy to follow. One suggestion from me is not to put too many things in a single section Introduction and Related Works''. The experiments are well conducted and analyzed. Some conclusions are demonstrated. But I have some concerns about experiments part. Check improvements'' part for details.

[Author Response · NeurIPS 2019]

Dear Reviewers of Submission #2112:

We thank for your reviews and constructive advices for this paper. Here is a response to your proposed issues.

**1. Experimental Setup for Fair Comparison.** In the experiment, we compare our method with NETRATE [2] and
KernelCascade [1] which aim at learning the hidden network among multi-dimensional markers in event sequences. We
use the similar settings as in [2, 1] in order to guarantee a fair comparison.

i) *Dataset and Preprocessing*: the generation of synthetic datasets is followed by [1]. Also, the MemeTracker dataset is
also used by [2, 1] and we adopt the similar preprocessing (filter out the top 500 sites). Moreover, we consider an extra
large synthetic dataset and another real-world dataset Weibo in order to test the performance on large-scale system.

ii) *Evaluation Protocols*: we follow [2, 1] and use the same metrics–precision, recall and F1-score–to measure the
accuracy of network reconstruction. For each method, we independently run the experiment five times and report the
averaged scores to avoid the accidental results.

iii) *Hyperparameter and Environment*: for each comparative method, we refer to the hyperparameter settings reported
in the paper and do some tuning to guarantee the optimal performance of them on each task. Finally, the achieved
performances of two baselines on MemeTracker are close to what are reported in their paper, and particularly, our
achieved F1 scores of them are even higher than what they reported. Besides, we implement each method on the same
hardcore environment (two Nvidia Tesla K80 with 12G memory) in order to guarantee the fairness of scalability test.

**2. Time and Space Complexity.** We divide the complexity analysis into feed-forward inference and back-propagation
training. The bottlenecks of these two parts in LANTERN respectively lie in the random-walk marker sampling and
learning for marker embedding. For the first bottleneck, we propose an equivalent efficient method that has linear
complexity for sequence length, so the time and space complexity of sampling one event sequence are both $O(T)$ where
$T$ denotes maximum length of sequence. For the second bottleneck, as discussed under Eqn. (2), we fix the probability
$p(m_j \in \mathcal{N}_i)$ in one epoch and update it when an epoch is finished. With such trick, we only need to update the marker
embedding for each marker once in an epoch, which requires $O(MT)$ in total, where $M$ is the number of markers.
Hence, in one epoch, the time and space complexity can be controlled within $O(MT)$, which is linear w.r.t marker
number and sequence length. By contrast, in previous works NETRATE [2] and KernelCascade [1], since they assume
each edge between two nodes (markers) entails a transmission function and do not sample neighbored nodes, the time
complexity would be $O(M^2T + MT^2)$ and space complexity would be $O(M^2T)$ in one epoch. Thus, our method
**reduces the quadratic complexity to linear one**. The scalability test in experiments verify this result.

**3. Hyperparameter Searching.** The hyperparameters used in our experiment are searched by coordinate descend. The
searching spaces for hyperparameters are as follows: learning rate $\alpha = [5e-7, 5e-6, 5e-5, 5e-4, 5e-3, 5e-$
$2, 5e-1]$, embedding dimension $D = [4, 8, 12, 16, 24]$, attention head number $L = [1, 2, 4, 8]$, regularization $\lambda =$
$[0.1, 0.01, 0.001, 0.0001]$, discount factor $\gamma = [0.8, 0.9, 0.99, 0.999]$, time embedding weight $\eta = [0.03, 0.3, 1, 3, 30]$.

**4. Reinterpretations for Scalability Results.** The scalability test results are presented in Fig. 5 in our paper, which
uses log-scale axis in order to reduce the range variation. To provide a more concrete comparison, we report the
running time for each method in Table 1. As you can see, when marker number goes to very large, e.g. $10^4$,
our method could be at least 10 times more efficient than NETRATE and at least 50 times more efficient than
KernelCascade. Importantly, we can see that as the marker number ranges from 100 to 100,000, the running time
of NETRATE and KernelCascade increases exponentially while our method LANTERN exhibits a linear increasing.
One can also see the trends from Fig. 5 in the paper, where the slope of LANTERN is **close to 1 (i.e., linear trend**
**in uniform axis)** while the slopes of two other methods are **close to 2 (i.e., quadratic trend in uniform axis)**.

Table 1: Comparison of running time for each method (the curves are shown in Fig .5)

| Seq. Length | 5 | | | | 25 | | | | 50 | | | |
|---|---|---|---|---|---|---|---|---|---|---|---|---|
| # of Marker | $10^2$ | $10^3$ | $10^4$ | $10^5$ | $10^2$ | $10^3$ | $10^4$ | $10^5$ | $10^2$ | $10^3$ | $10^4$ | $10^5$ |
| NETRATE | 0.2 | 2.6 | 1018.3 | 72142.1 | 5.8 | 463.4 | 25075.5 | - | 23.5 | 1853.7 | 100300.5 | - |
| KernelCascade | 0.4 | 7.3 | 7938.6 | - | 10.4 | 183.2 | 108465.8 | - | 41.6 | 3333.0 | 433863.5 | - |
| LANTERN | 0.3 | 2.2 | 24.3 | 218.3 | 22.3 | 167.1 | 2832.4 | 12211.2 | 81.4 | 672.6 | 7333.8 | 65135.6 |

**5. Revisions for Confusing Terminology.** We apologize for misusing the 'causal' term to describe our hidden network
among markers. The original motivation of using 'causal graph' is based on the causal-effect relations between two
marked events in a sequence, but we ignore the fact that in a rigorous sense a causal graph is to represent the causal
relations among different random variables. We would replace it as 'hidden influence network' in the final version.

# References

[1] N. Du, L. Song, A. J. Smola, and M. Yuan. Learning networks of heterogeneous influence. In *NIPS*, pages 2789–2797, 2012.

[2] M. Gomez-Rodriguez, D. Balduzzi, and B. Schölkopf. Uncovering the temporal dynamics of diffusion networks. In *ICML*,
pages 561–568, 2011.


[Meta-Review · NeurIPS 2019]

This paper received two strongly favorable reviews and one negative review. The negative review is about writing, and I do share the concern. For example, I'm a lost at the motivation behind the Marker Generation process as described underneath Theorem 1 (first bullet). No intuition is given on why it cannot stop when m_i \in \Mcal_k. The paper does say "Note that it is possible that an event is caused by more than one events before, and we only consider the first causal parent as the true causal parent [11, 9, 8]." But that doesn't forbid a marker from recurring. In fact, Figure 2 does allow m_4 to occur twice. Please clarify it. Theorem 2 looks quite interesting which reduces the complexity by an order of magnitude. But one needs to be convinced that the Marker Generation process makes sense in the first place. Overall the technical merit is solid, allowing latent process from high-dimensional event sequences to be learned. However, I urge the authors take the comment of Reviewer #1 in to serious account when preparing the camera-ready version.